# Phylogenetic Relationships and Evolution of the Genus *Eganvirus* (186-Type) *Yersinia pestis* Bacteriophages

**DOI:** 10.3390/v16050748

**Published:** 2024-05-08

**Authors:** Jin Guo, Youhong Zhong, Yiting Wang, Pan Liu, Haixiao Jin, Yumeng Wang, Liyuan Shi, Peng Wang, Wei Li

**Affiliations:** 1National Institute for Communicable Disease Control and Prevention, China CDC, Changping, Beijing 102206, China; guojin1523215@163.com (J.G.); wangyiting@icdc.cn (Y.W.); 18779114353@163.com (H.J.); wangyumeng@icdc.cn (Y.W.); 2National Key Laboratory of Intelligent Tracking and Forecasting for Infectious Diseases, Beijing 102206, China; 3Yunnan Institute for Endemic Disease Control and Prevention, Dali 671000, China; zyhong520@126.com (Y.Z.); liup0506@163.com (P.L.); 15087261364@163.com (L.S.); 4Yunnan Provincial Key Laboratory for Zoonosis Control and Prevention, Dali 671000, China

**Keywords:** *Yersinia pestis*, 186-type bacteriophages, phylogenetic analysis, genus *Eganvirus*, *cI*, *apl* and *cII* genes, integrase

## Abstract

Plague is an endemic infectious disease caused by *Yersinia pestis*. In this study, we isolated fourteen phages with similar sequence arrangements to phage 186; these phages exhibited different lytic abilities in Enterobacteriaceae strains. To illustrate the phylogenetic relationships and evolutionary relationships between previously designated 186-type phages, we analysed the complete sequences and important genes of the phages, including whole-genome average nucleotide identity (ANI) and collinearity comparison, evolutionary analysis of four conserved structural genes (*V*, *T*, *R*, and *Q* genes), and analysis of the regulatory genes (*cI*, *apl,* and *cII*) and integrase gene (*int*). Phylogenetic analysis revealed that thirteen of the newly isolated phages belong to the genus *Eganvirus* and one belongs to the genus *Felsduovirus* in the family *Peduoviridae*, and these *Eganvirus* phages can be roughly clustered into three subgroups. The topological relationships exhibited by the whole-genome and structural genes seemed similar and stable, while the regulatory genes presented different topological relationships with the structural genes, and these results indicated that there was some homologous recombination in the regulatory genes. These newly isolated 186-type phages were mostly isolated from dogs, suggesting that the resistance of Canidae to *Y. pestis* infection may be related to the wide distribution of phages with lytic capability.

## 1. Introduction

Plague, caused by the Gram-negative bacterial pathogen *Yersinia pestis*, is a notorious zoonotic disease. Plague primarily infects humans through flea bites, mainly causing bubonic plague with swollen lymph nodes. Other means of infection include contacting blood or inhaling aerosol droplets with *Y. pestis*, which can cause pneumonic plague [1]. Currently, plague occurs in some countries, including China [2,3], the Democratic Republic of the Congo, Uganda, Madagascar, and the United States [4].

To date, public health measures and effective antibiotic treatments have significantly reduced plague infections worldwide. However, the disease remains ineradicable, and endemic natural plague foci persist across Africa, Asia, and the Americas [5]. Multidrug resistance in *Y. pestis* strains, mediated by conjugative plasmids, has been reported in Madagascar [6,7]. Furthermore, novel streptomycin resistance mechanisms, including a *rpsL* gene (ribosomal protein S12) mutation at amino acid 43 (K43R), have been identified in strains from China and Madagascar [8,9]. Additionally, a case of plasmid-mediated doxycycline resistance was documented in a *Y. pestis* strain isolated from a rat in Madagascar in 1998 [10].

Bacteriophages were first explored for their ability to treat plague cases in 1925 when Dr. d’Herelle attempted to treat four bubonic plague cases using *Y. pestis* phage (YpsP-G) [11]. In addition, phage therapy could be a new potential strategy for treating plague infection or as a biocontrol method for eliminating the endemics of animal plague in natural plague foci [12], such as the genus *Gaprivervirus Y. pestis* phages with broad host range characteristics, vB_YpeM_MHS112 (OP750247) and vB_YpeM_GMS130 (OP750248) [13].

In fact, phages shape the composition and evolution of bacterial communities in nature and are, therefore, important in ecosystem cycles [14]. Phage communities affect the composition of infected bacteria through various mechanisms, including lysogenic conversion, transduction, and host gene disruption, when they integrate into the bacterial genome as prophages [14].

P2-like phages are a group of related temperate phages that are commonly found in γ-proteobacteria. Generally, the genome architecture of P2-like phages seems to be more conserved than that of majority phages from other families or classes [15]. Structural genes are more conserved than transcriptional switches in P2-like phages, with the latter controlling the lytic versus the lysogenic growth cycle [15]. Previous studies have distinguished P2-like phages into two major types, i.e., p2-type phages and 186-type phages [15,16,17]. P2-type phages and 186-type phages are morphologically indistinguishable, both having an icosahedral capsid and a contractile tail. However, the genes and patterns of regulatory genes of the 186-type phages are quite different from those of p2-type phages, and the 186-type phages appear more complex and contain the additional gene *cII* [15]. According to the latest classification of bacteriophages by the International Committee on Taxonomy of Viruses (ICTV), phage 186-type phages were found in the genus *Eganvirus* in subfamily *Peduovirinae*, while P2-type phages were found in the genus *Peduovirus* [18].

Our team isolated several wild-type *Y. pestis* phages, including previously called T4-like phages [12,19] and P2-like phages [20]. Through genomic analysis, we identified 14 *Y. pestis* phages that are highly homologous to phage 186 in Yunnan plague natural foci. In this study, the morphological characteristics and host range of these new phages were observed, and we further conducted comparative genomic analysis, including whole-genome average nucleotide identity (ANI) and collinearity comparison. In addition, we performed a phylogenetic analysis of four capsid-related structural genes as in the previous literature [15], namely, the capsid scaffold gene (*V*), the major capsid precursor gene (*T*), the small terminase subunit gene (*R*), and the capsid completion gene (*Q*), which correspond to the structural genes *O*, *N*, *M*, and *L* in P2-type phages. We also compared the phylogenetic relationships of regulatory genes associated with 186-type phages, including the *cI*, *apl*, *int*, and *cII* genes, which are crucial for lysogeny/lysis of 186-type phages.

## 2. Materials and Methods

### 2.1. Isolation and Identification of Bacteriophages

Fourteen wild-type phages were isolated in 2016 and 2018 from the anal swabs of dogs (*Pug*, *Chinese rural*, *and Pekingese*) and the caecum of small rodents (*Apodemus chevrieri*) in two natural plague foci in Yunnan, China (Table 1). All samples were soaked in PBS-glycerol (1:1, *v*/*v*), centrifuged at 9000× *g* for 10 min, and then filtered through a 0.22 μm pore size membrane filter (polyethersulfone membrane, hydrophilic, Millex^®^-GP filter). The *Y. pestis* vaccine strain EV76 was used as an indicator host. The above filtrate was incubated with 200 μL of the EV76 strain in LB broth (OD600 ≈ 0.4) and shaken (220 rpm) at 28 °C for 24 h. Next, the phages were filtered through a 0.22 μm filter and added to 500 μL of *Y. pestis* EV76 culture until the liquid became clear. Single phage plaques were separated and purified by a double agar plaque assay according to the literature [13,20].

### 2.2. Transmission Electron Microscopy

The purified phage preparations were spotted onto 400 mesh carbon-coated grids, negatively stained with 2% phosphotungstic acid, and then visualised using a Hitachi HT7700 80-kV transmission electron microscope (Tokyo, Japan) with a Gatan 832.10 W CCD camera (Gatan, Pleasanton, CA, USA) operating in Gatan Digital Micrograph software [21]. The diameters of the heads and lengths of the tails were determined from electron micrographs. At least three well-defined micrographs were used to measure the sizes of the phages, and the averaged measurements are listed in Table 1.

### 2.3. Host Range Analysis

The bacterial strains used for the host range assay are listed in Appendix A. A total of 114 *Enterobacteriaceae* strains (15 genera) were selected, including the genera *Yersinia*, *Escherichia*, *Salmonella*, *Shigella*, *Cronobacter*, *Klebsiella*, *Enterobacter*, *Enterococcus*, *Cronobacter*, *Serratia*, *Proteus*, *Providencia*, *Morganella*, *Citrobacter*, and *Pantoea*. A spot testing assay was used for host range analysis [13]. Briefly, 400 μL of bacterial culture liquid was added to 50 °C molten LB semisolid agar, mixed and poured into the prepared 36 grids of nutrient solid plates to make a double-layer plate. After solidification of the top agar matrix, 4 μL of phage filtrate was dropped on top agar and incubated at 37 °C (for Enterobacteriaceae bacteria) or 28 °C (for EV76 and other *Y. pestis* strains) for 12–36 h to observe the plaque results. The host range was visualised through Tree Visualisation by One Table (TvBOT, https://www.chiplot.online/tvbot.html, v 2.6) (accessed on 22 December 2023), as illustrated in a heatmap [22].

### 2.4. Bacteriophage DNA Extraction and Amplification

Phages were propagated in 500 μL of LB medium supplemented with *Y. pestis* EV76 strain culture with shaking (220 rpm) overnight at 28 °C, and phage DNA was extracted using an ABigen corporation Lambda phage genomic DNA extraction kit (ABigen Corp., Beijing, China) according to previous methods [13]. Soapnuke (v2.0.5) was used for quality control.

### 2.5. Phage Genome Sequencing, Assembly, Annotation, and Comparison

The whole genomes of fourteen 186-related phages were sequenced by a commercial sequencing company (MAGIGENE, Shenzhen, China). BWA (v0.7.17) software was used to align and clean the reads associated with the host genome to remove the host genomic sequence. Megahit software was used for de novo assembly (v1.1.2 SCIME) [23]. MetaGeneMark (v3.38) software was used to predict the genes of contigs in the target genome (sequences with nucleic acid lengths less than 150 bp were filtered) [24,25]. Two methods were used to annotate gene function, i.e., BLASTP (v2.9.0+) software was used to compare the gene protein sequence with the virus sequence in the UniProtKB/Swiss-Prot database (ViralZone, reviewed Proein, https://viralzone.expasy.org/) (accessed on 15 August 2023) to obtain functional information for the genes [26]. Second, the putative gene function was predicted using the RAST tool (https://rast.nmpdr.org/, accessed on 22 August 2023) [27] and supplemented by analyses performed with PHASTER (https://phaster.ca/, accessed on 12 January 2024) [28,29] and the Galaxy web service (https://usegalaxy.org/, accessed on 12 January 2024) [30] to enhance the accuracy of the presumed gene function prediction. The genomic sequence comparisons among phages were conducted using Easyfig software [31]. FastANI software was used to conduct pairwise comparisons of phage genomes at the nucleotide level by calculating their average nucleotide identity (ANI) [32]. The values of the ANI multiplied by the coverage were used in homological analysis.

### 2.6. Phylogenetic Relationship Analysis

A total of 62 genomes of P2-like phages or prophages were used to analyse phylogenetic relationships in this study, including 14 newly isolated 186-related phages in this study (Table 1), P2-like phages, or prophages previously used in phylogenetic analysis in the literature [15], and all seven phage strains in the genus of *Eganvirus* (186, EtG, PsP3, SW9, BIS20, SI22, and SEN1) obtained from the National Center for Biotechnology Information (NCBI) database (Appendix A). Another three phages (vB_EcoM-613R3, vB_Sal_PHB48, 3625_26581) highly homologous to phage 186 were also included in this study. Our newly isolated phages were identified as 186-type phages that met the following two conditions: (1) classified as a member of the *Peduovirinae* subfamily; and (2) had four late structure genes of phage 186 (*V*, *T*, *R,* and *Q* genes) and complete regulatory region gene arrangements (*cI*, *apl*, *cII,* and *int* genes). In addition, to illustrate the phylogenetic relationships and population positions of these 186-type phages (prophages), the phylogenetic relationships of 15 *Peduovirus* phages (including phages the P2, L-413C, P2 Hy dis, Wφ, vB_YpM_22, vB_YpM_46, vB_YpM_50, etc.) were also investigated via genome and structure protein phylogenetic relationship analysis (Appendix A).

Phylogenetic trees were constructed using IQ-TREE (v2.2.2.6) software [33] and the maximum likelihood (ML) method [34] based on the inferred amino acid sequences from four structural genes as mentioned in the previous literature [15] (notes: *V*, *T*, *R,* and *Q* genes analysed together) and four regulatory genes (*int*, *cI*, *apl,* and *cII* genes). A phylogenetic tree of the full-length genome was constructed using the Bayesian inference (BI) method and OrthoFinder software [35]. To elucidate phylogenetic relationships, we utilised PhyloSuite software (v1.2.2) [36], which employs the MAFFT (v7.313) [37] function for aligning protein sequences. The concatenated sequences of structural genes were then analysed phylogenetically using IQ-TREE software with default parameters (-m TEST -nt AUTO -bb 1000) [33]. The constructed phylogenetic trees were then imported into tvBOT for further enhancement and visualisation [22].

The consistency of major clusters was verified through homogeneity tests, and branches with less than 50% support were collapsed. The stability of the phylogenetic tree was evaluated for robustness using 1000 bootstrap replicates, and the corresponding values were indicated on the branches.

### 2.7. Accession Numbers

The whole-genome sequences of the newly isolated 186-related phages in this study have been deposited in GenBank, and the detailed accession numbers can be found in Table 1.

## 3. Results

### 3.1. Morphology of the Y. pestis Phages of the Genus Eganvirus

Electron microscopic observations indicate that the phages of the genus *Eganvirus Y. pestis* have a typical icosahedral head and long contractile tail. Thus, based on these morphologic characteristics, these phages have *Caudoviricetes* characteristics (Figure 1). These newly isolated *Eganvirus* phage genera exhibited roughly consistent head and tail sizes. The average head diameter was 71 ± 8 nm, and the average tail length was 156 ± 16 nm (Table 1). However, the size of phage MLG42 was notably greater than that of its counterpart (Figure 1).

### 3.2. Host Range Determination

Host range analysis was conducted for 13 of the 14 newly isolated *Y. pestis* phages (MDG94 was not recovered successfully) in this study. These phage isolates can infect twenty wild-type *Y. pestis* strains (biovars including Antiqua, Mediaevalis, Orientalis, and Microtus) in China. Host lysis experiments revealed that these phages presented different infecting host profiles; for example, MDG45 could not lyse any of the tested strains, whereas other phages were sensitive to at least one host strain (Figure 2, Appendix A). MLG42, MHG7, and MHG48 could completely lyse some *Yersinia pseudotuberculosis*, while MHG96, MHG101, MHG54, MHG7, MHG48, and GCS667 could incompletely lyse at least one *Y. pseudotuberculosis*; phage GCS667 incompletely lysed six of seven tested *Y. pseudotuberculosis*; phages MLG42, MHG7, and MHG48 could completely lyse at least one *Yersinia enterocolitica*, whereas MHG54 and GCS667 incompletely lysed *Y. enterocolitica*. In addition, phage MHG39 was the only phage that could lyse *Yersinia kristensenii.*

Regarding the lysing panels of *Escherichia coli*, phage MHG39 was particularly notable and could lyse five *E. coli* strains, followed by MHG48 lysing two strains, and MHG51, MHG101, MLG42, and MHG54 each lysing one strain; however, their lytic profiles were different. Both MHG39 and MHG54 could lyse *Shigella flexneri*, and MHG55, MHG75, and MHG39 could lyse *Shigella boydii*. *Shigella sonnei* could be lysed by more than half of the 186-related phages. Additionally, *Cronobacter malonate-positive strains* could be partially lysed by MHG54 and MHG48, and *Enterobacter cloacae* could be completely lysed by GCS667. *Salmonella blegdam* was lysed completely or partially by MHG39, MHG54 and MHG48.

### 3.3. Homology and Average Nucleotide Identity (ANI)

Except for phage MLG42, the genomic sequences of the newly isolated *Y. pestis* phages were homologous to those of the genus *Eganvirus*. The ANI of these *Y. pestis* phages ranged from 97.359%–99.99%. In total, the genus *Eganvirus* phages were considerably phylogenetically different from the genus *Peduovirus* phages, as phages previously called P2-like phages (value of ANI multiple coverage ranging from 25% to 36%) (Figure 3, Appendix A).

### 3.4. Genomic Collinearity Comparison

Collinearity analysis indicated that the genome sequences of phages from the newly isolated genus *Eganvirus* (MDG45 and GCS667 as representatives) were similar to those of *E. coli* phage 186 or ETG (Figure 4A,B) [38]. These newly isolated *Y. pestis* phages presented highly similar gene arrangements; however, some detailed differences existed when comparing whole-genome sequences (Appendix A). MHG101, MHG48, MHG54, MHG7, MHG39, and MHG51 exhibited almost the same gene arrangements, while MHG75, MHG96, MHG38, MDG45, and MDG94 exhibited similar arrangements of a second type; furthermore, GCS667 and MHG55 had more differences in their genomic sequences when compared to the other genomes (Appendix A).

The phage MLG42 belongs to the genus *Felsduovirus*. When MLG42 was compared with P2-like phages (or prophages) in the NCBI database, MLG42 exhibited more coverage and similarity with the phages Fels-2 and SopEφ (Figure 4C), and the ANI values ranged from 89.84% to 93.93% within Fels-2, SopEφ, 3625_26581, φSEN2 (prophage), and MLG42 (Appendix A).

### 3.5. Phylogenetic Relationship between Genomic Proteins and Four Conserved Structural Proteins

These 13 newly isolated *Eganvirus* phages clustered together with phages from other sequenced genera of *Eganvirus* (phage 186, EtG, PsP3, SW9, and BIS20), but it was obvious that the phages of these genera could be roughly clustered into three subgroups based on their genomic sequences (Figure 5A). The phage MLG42 was clustered into the genus *Felsduovirus* together with Fels-2, SopEφ, 3625_26581, and φSEN2 (Figure 5A).

P2-like phages possess conserved late structural genes. A total of 48 P2-like phages or prophages, most of which were described in the previous literature [15,39], along with 13 phages of the newly isolated genus *Eganvirus Y. pestis* (Table 1) and 1 phage of the genus *Felsduovirus Y. pestis* (MLG42), were selected for the comparison of four conserved structural proteins (V, T, R, and Q for 186-type phages, corresponding to O, N, M, and L for the genus *Peduovirus*). The phylogenetic relationships of these previously identified P2-like phages are illustrated in Figure 5B. One notable feature of these relationships was that the phylogenetic topology of the four structural genes (Figure 5B) was analogous to the topology of the whole genomic sequences (Figure 5A), indicating the exclusion of diversity and discrimination scales in their differentiation. 

### 3.6. Regulatory Region Genes

A comparison of the topological relationships among the *apl*, *cI*, and *int* genes revealed distinct differences (Appendix A). For example, one subgroup of *Eganvirus* phages (depicted in deep blue in Figure 6) appeared to be closer to phages in the genus *Felsduovirus* when phylogenetic relationships were based on the *cI* or *int* genes. Moreover, another subgroup 2 (shown in Figure 5) of *Eganvirus* phages (light blue in Appendix A) seemed to have a closer relationship with the phage *Felsduovirus*.

The most significant feature of genus *Peduovirus* phages and genus *Eganvirus* phages was that the latter contains the additional gene *cII.* Excluding the diversity and discrimination scales in their differentiation, one interesting observation about the phylogenetic topology of the CII protein (Figure 6) was that it was unexpectedly analogous to the topology of the four structural proteins (Appendix A). That is, phages belonging to the same genus also had identical sorting of these four constructed proteins (Figure 5 and Figure 6).

Among the 13 newly isolated *Eganvirus* phages, the *int*, *cI,* and *cII* genes are highly homologous. However, surprisingly, the APL protein of GCS667 showed a different phylogenetic relationship with the other phages, as it clustered more closely with phage 186, EtG, and two prophages (φESP and ΦKPN). Corresponding clustering results based on genomic sequences also revealed a similar result (Figure 4 and Appendix A).

## 4. Discussion

### 4.1. Nomenclature of Y. pestis Phages

Many phages can lyse *Y. pestis*, including lytic phages previously called *Podoviridae* phages (such as Pokrovskaya [40], A1122 [41], and Yep-phi [42]), as well as eight recently reported new phages with *Podoviridae* characteristics (YpEc56, YpEc56D, YpEc57, YpEe58, YpEc1, YpEc2, YpEc11, and YpYeO9) [43]. Because several such phages have specific lytic effects on *Y. pestis*, some are routinely used for the diagnosis of this disease. However, some such phages also exhibit lytic effects on some *Y. pseudotuberculosis* strains [11].

Another type of *Yersinia* lytic phage is the T4-like phage, and T4-like phages can generally infect a broad range of hosts [44], sometimes spanning different bacterial genera [14]. T4-like *Yersinia* phages with a wide host spectrum are common [45], such as fD1 [46], JC221 [21], YpsP-PST (KF208315.1) [40], fPS-2 (LR215722), fPS-65 (LR215724), and fPS-90 (LR215723) [19], as well as two *Gaprivervirus* genus *Y. pestis* phages (MHS112 and GMS130) [13]. Some of these T4-like phages even appeared to reproduce much more efficiently in *E. coli* strains than in *Y. pestis* [13,46].

Many P2-like *Yersinia* phages, such as L-413C (NC_004745) [38], vB_YpM_22, vB_YpM_46 vB_YpM_50 [47], and HQ103 (MZ420230.2) [20], have been isolated from various environmental backgrounds [48]. The *Yersinia* phage L-413C was previously reported to lyse *E. coli* and was once suggested to be an *E. coli* phage [15,38]. In fact, these phages are independent species, and phage names labelled with indicator hosts provide only some background for understanding their phenotypic characteristics.

### 4.2. Diversity of Phages from the Genus Eganvirus

The newly isolated *Y. pestis* 186-type phages exhibit a high degree of genomic consistency. This consistency was not only observed in terms of genomic arrangement (99–100%), but also in high homology to conserved structural genes. One interesting phenomenon is that when the *apl* and *cI* genes were compared, the phage GCS667 clustered apart from other *Eganvirus* phages, and instead GCS667 clustered with the phages EtG and φESP (prophage). In contrast GCS667 clustered with other newly isolated *Eganvirus* phages when the *int*, *cII,* and structural genes were compared (Figure 5 and Figure 6, Appendix A).

The total genus *Eganvirus* phages presented a certain degree of consistency (Appendix A), however, they exhibited clear divergence, even though the genomic sequence or the key regulatory genes could differentiate at least two to three subgroups (Figure 5A). This scenario indicated that there were some differences in the genus *Eganvirus* phages and genus *Eganvirus* phages that can be subdivided into more varieties in the future.

### 4.3. Phylogenetic Relationship between Regulatory Genes and Structural Genes

In this study, we analysed 14 newly isolated phages, 13 of which belong to the genus *Eganvirus*, while one (MLG42) belongs to the genus *Felsduovirus*. These phages were compared with other members of the genus *Eganvirus* and further compared with P2-like phages using previous methods [15] to illustrate their phylogenetic relationships based on four conserved structural genes or key regulatory genes. The phages in this comparison included the genera *Entnonagintavirus*, *Felsduovirus*, *Longwoodvirus*, *Bielevirus*, *Hpunavirus*, and *Irtavirus*, as well as the more common *Peduovirus* (only representative P2-like phages were included).

One obvious convergence among these 186-type phages could be observed when these phages were grouped together based on whole-genome sequences or four relatively conserved structural genes (*V*, *T*, *R,* and *Q*). The phylogenetic trees based on the two methods revealed a general concordance in the phylogenetic arrangement, with only subtle shifts observed. This scenario indicates that the general genomic diversity of these phages could also be represented by the variation in the four relatively structural genes.

The CII protein, a member of the helix-turn-helix family of transcriptional activators, is pivotal for establishing lysogeny by activating a promoter (pE) that initiates the expression of the CI immune repressor, which is essential for efficient lysogenic establishment [49]. If we disregard the diversity discrimination scales in their differentiation, the phylogenetic topology of the CII protein was more closely analogous to that of the four conserved structural genes.

Despite these similarities, interesting phylogenetic discrepancies could be observed, such as the relationships illustrated based on the *cI*, *apl,* and *int* genes (Figure 6), which differ from those based on the evolution of structural genes or genomic diversity subtyping (Figure 5). Such phenomena indicate that the evolution of the *cI*, *apl,* and *int* genes differs from that of the structural genes, and this distinction is likely attributed to homologous recombination [15].

Phage MLG42, which was isolated from a domestic dog anal swab, exhibited marked differences from the other phages. The distinct nature of the phage MLG42 can be reflected through many features, such as the phylogenetic relationships of regulatory and structural genes, attachment site *attP*, genome sequence, and even electron microscopy morphology (Figure 1). Therefore, we concluded that MLG42 is a new member of the genus *Felsduovirus*.

### 4.4. Phages Lysing Y. pestis Are Widely Distributed in the Environment

Yunnan is considered the origin of the Third World plague pandemic in the middle of the 19th century [50]. All *Eganvirus* phages in this study were isolated from Yunnan Province. An overall investigation of phages in indicator animals (dogs and cats) was performed from 2015–2018 in Yunnan Province [51]. The investigation covered 41 villages in 26 towns located in 10 counties or city districts and involved a total of 1003 dogs. Phages were isolated from anal swabs of these indicator animals, using *Y. pestis* EV76 as the host indicator. A total of 102 phages lysing *Y. pestis* were isolated from dogs, and these phages presented different polymorphisms based on electronical microscopic morphology; some were identified as P2-like phages (such as GMG73) or T4-like phages [13]. These results indicate that phages that can lyse *Y. pestis* are widely distributed in the environment. In addition, most phages in this study were isolated from Menghai County. Menghai County belongs to the *Rattus flavipectus* plague focus, and one bubonic plague case with animal plague epizootics was confirmed in Menghai County in 2020. However, due to a lack of opportunities, the phage-carrying situation in dogs was not investigated until 2020.

### 4.5. The Significance of Phages Isolated from Dogs

In this study, many *Eganvirus* phages were isolated from dog samples. Canidae (such as wolves, dogs, coyotes, and foxes) are resistant to plague disease [52]. Although coyotes and shepherd dogs are frequently infected with *Y. pestis* through the consumption of plague-infected prey, dogs or coyotes are thought to be resistant to *Y. pestis* infection, and these animals generate plague-protective F1 antibodies without developing disease symptoms [53]. Indeed, dogs are believed to be indicator animals for surveillance of plague zoonosis in plague foci [54].

Recent research on LcrV (the needle cap protein of the *Y. pestis* type III secretion system) and the N-formylpeptide receptor (FPR1) revealed that dogs lack susceptibility to *Y. pestis* [55]. FPR1 is a plague receptor on immune cells in both humans and mice, and *Y. pestis* binds to the FPR1 receptor to promote the translocation of bacterial effectors through a type III secretion system, which selectively destroys immune cells in humans. This mechanism enables *Y. pestis* to reproduce in the bloodstream and be transmitted to new human or mouse hosts through fleabites and mass propagation [55]. Because Canidae have lost *Fpr1* and the ability to respond to N-formyl peptides, it can be inferred that plague resistance in Canidae may be related to the loss of FPR1 in these species [55].

Many *Yersinia* phages have been isolated from dogs by our team, as mentioned above. These phages do not rely solely on *Y. pestis* as their host bacteria; instead, they can propagate through a broader range of *Enterobacteriales* species and form an ecological barrier against *Y. pestis* invasion. This scenario maybe could provide another explanation for the resistance of dogs or coyotes to *Y. pestis* infection, potentially related to the widespread distribution of phages with the ability to lyse *Y. pestis*.

## Figures and Tables

**Figure 1 viruses-16-00748-f001:**
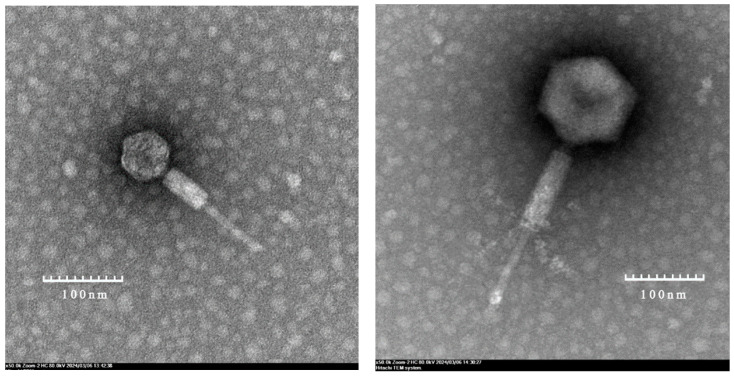
Transmission electron microscopy images of the morphology of the MHG48 (**left**) and MLG42 (**right**) phages.

**Figure 2 viruses-16-00748-f002:**
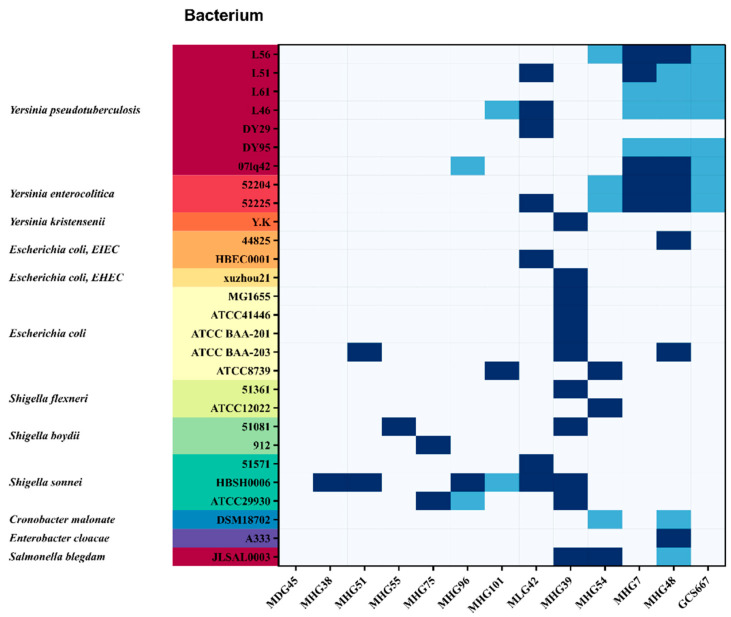
Host range of phages from the newly isolated genus *Eganvirus Y. pestis* phages. Notes: The bacteria used for host range tests are shown with different colours, and only the bacteria exhibiting lysis phenomena are listed on the vertical axis. The bacteria without lysis among the 114 host strains tested are not shown (detailed results are listed in Appendix A). Dark blue indicates complete lysis (clear circular plaques), light blue indicates incomplete lysis (faint and blurry circular lysis or dispersed lytic points), and colourless indicates no lysis.

**Figure 3 viruses-16-00748-f003:**
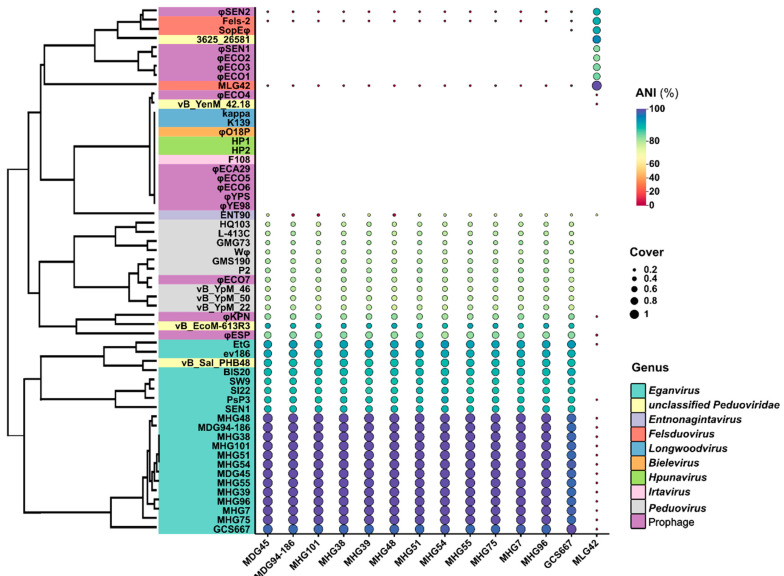
Homology and average nucleotide identity (ANI). Notes: The line length of classification on left do not reflect the discrimination degrees.

**Figure 4 viruses-16-00748-f004:**
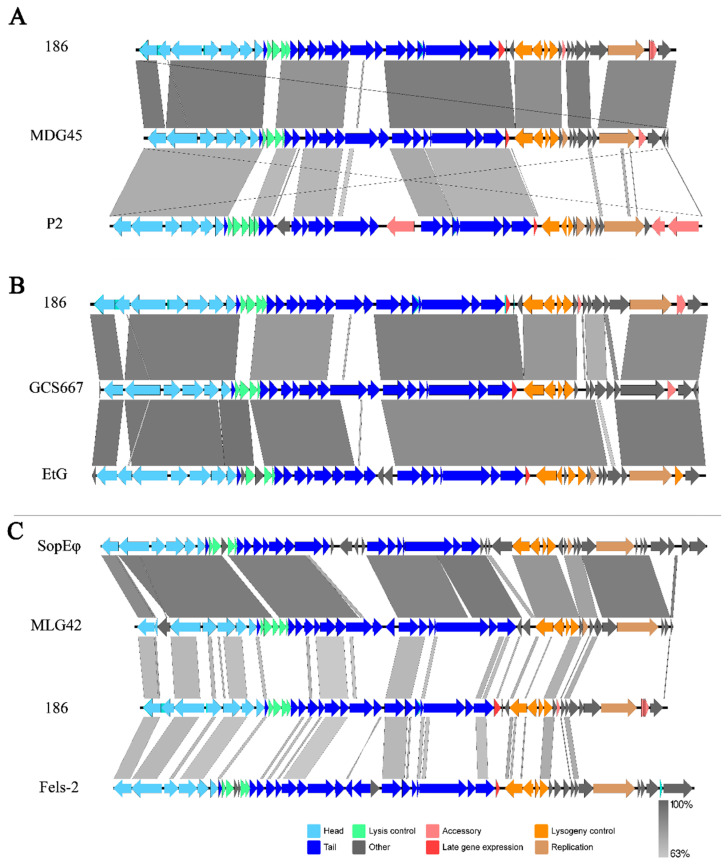
Genomic collinearity comparison of the *Eganvirus* phage MDG45 (**A**), GCS667 (**B**), and *Felsduovirus* phage MLG42 (**C**).

**Figure 5 viruses-16-00748-f005:**
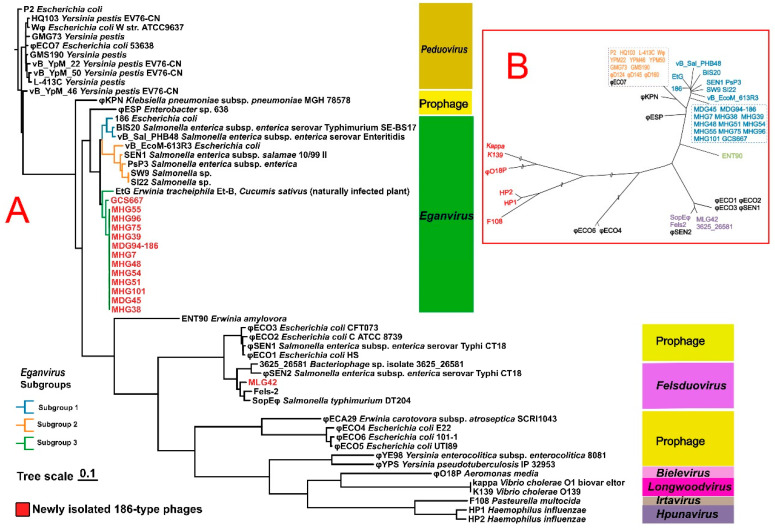
Phylogenetic relationship of genomic proteins (**A**) and four conserved structural proteins (**B**).

**Figure 6 viruses-16-00748-f006:**
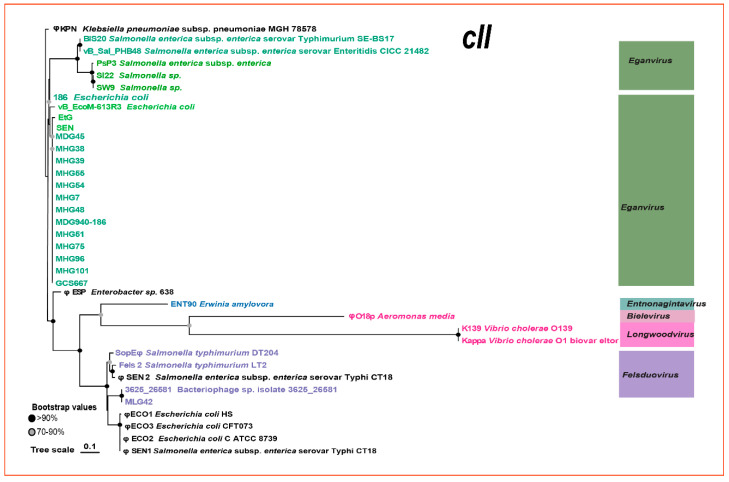
Phylogenetic relationship of the inferred amino acid sequences of the *cII* gene.

**Table 1 viruses-16-00748-t001:** Newly isolated 186-type phage information in this study.

Name	Host/Sources	Natural Plague Foci	GC (%)	Sequence Size (bp)	Head Size (nm)	Tail Length (nm)	Sequence Accession No.
MDG45	*Pug dog*, anal swab	F	53.7	29,697	78.8	157.6	PP516624
MHG38	*Pug dog*, anal swab	F	53.8	29,826	72.7	169.7	PP516625
MHG51	*Pug dog*, anal swab	F	53.7	29,687	78.8	163.6	PP537603
MHG55	*Chinese rural dog*, anal swab	F	53.7	29,758	90.0	133.0	PP537604
MHG75	*Chinese rural dog*, anal swab	F	53.8	29,011	72.7	175.8	PP537605
MHG96	*Chinese rural dog*, anal swab	F	53.8	29,902	72.7	169.7	PP537606
MHG101	*Chinese rural dog*, anal swab	F	53.8	29,821	66.7	187.9	PP537607
MHG39	*Chinese rural dog*, anal swab	F	53.7	29,627	60.6	157.6	PP599994
MHG54	*Pekingese dog*, anal swab	F	53.7	29,697	60.0	140.0	PP599995
MHG7	*Chinese rural dog*, anal swab	F	53.7	29,697	66.7	136.4	PP599996
MHG48	*Pug dog*, anal swab	F	53.7	29,741	67.6	145.1	PP599997
GCS667	*Apodemus chevrieri*, *Caecum*	E	53.7	29,704	69.6	150.5	PP475179
MDG94-186	*Pug dog*, anal swab	F	53.7	29,569	67.6	145.1	PP516623
MLG42 *	*Chinese rural dog*, anal swab	F	54.8	31,231	128.4	234.3	PP599993

The natural plague foci included F: *Rattus flavipectus*, which is in Yunnan–Guangdong–Fujian provinces; E: *Apodemus chevrieri-Eothenomys miletus*, which is in the highlands of northwestern Yunnan Province; and *: MLG42, which belongs to the genus *Felsduovirus*.

## Data Availability

Data are contained within the article and Appendix A.

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
