# Peer review of "Phylogenetic Relationships and Evolution of the Genus Eganvirus (186-Type) Yersinia pestis Bacteriophages"

_viruses, 2024, doi:10.3390/v16050748_

Round 1
Reviewer 1 Report
Comments and Suggestions for Authors
The article “The Phylogenetic Relationships and Evolution of the genus Eganvirus (186-type) Yersinia pestis Bacteriophages” by Jin Guo reports the results of general biological and bioinformatic analysis of several newly isolated P2-related bacteriophages infecting Yersinia pestis. The manuscript contains an interesting phylogenetic analysis and an informative discussion regarding evolutionary aspects of the phage 186 related phages. The manuscript is well written and contains useful illustrations. However, there are a few caveats that need to be taken into account, as well as general concerns about the quality of some microscopy data and the visualization of the results.
Table 1. Some phage sizes look strange. They are too big for the P2-like phages. Please clarify.
Section 2.5. Phage annotation can be significantly improved using HMM searches. For example, HHpred. Please, use this server to check and improve annotations.
Lines 145-149. There is a special tool called VIRIDIC, approved by the ICTV. You could use it to infer the taxonomy of phages, which will be accepted officially. Also, you should indicate whether you took into account the coverage when calculated the ANI/ It is important to classify the phages correctly (Turner et al., 2021. “A Roadmap for Genome-Based Phage Taxonomy”)
Line 167 - please clarify the IQ-TREE command line parameters.
Line 173 - please clarify the MAFFT version and parameters used.
Line 173 and elsewhere - As far as I understand, you used the concatenated alignments. If it is so, please replace “The aligned tandem structural genes” all “ aligned together” to “concatenated alignments” or “concatenated sequences”.
Line 184 - There are no accession numbers either in section 2.8 or in table 1. Please provide accessions.
Figure 3. Instead of this figure you could provide the VIRIDIC heatmap. And in any case, you must take into account the coverage, to classify the phages correctly. Obviously, this doesn't consider the coverage.
Figure 4 - Please provide taxonomy in labels or captions. It is difficult to understand why you have chosen these phages. Please explain this.
Figure 5. Please improve the resolution. Please italicize all necessary taxonomic names.
Figure 6. Please improve the resolution. It is really bad now. Please italicize all necessary taxonomic names.
Line 326 - Please clarify what you mean by “significantly”. How much similarity of genomes is this?
Lines 393-398 - I would remove this hypothesis. This is interesting, but requires more justification and discussion.
Reviewer 2 Report
Comments and Suggestions for Authors
Dramatic changes in viral taxonomy with substantial elevation of previously regarded taxa have promoted an effort on the refinement of viral (including bacteriophages) attribution of families, genera and species. The reviewed paper of Guo et al. is devoted to the clarification of taxonomic positioning of P2-like phages of Yersinia. The laborious bioinformatic work results in the attribution of most studied phages to genus Eganvirus with further potential subdivision into three subclusters. The idea of the work is significant enough, but the results are presented in an arguable manner. Therefore, I'd recommend a substantial revision in visualization and presentation of the data.
1. The difference in the EM-revealed size of phage particles (eg. 60 to 90 nm capsid) is too big for the representatives of the same genus;
2. The accession numbers of the genomes are not presented in Table 1 or Data availability statement;
3. The choice of marker genes for phylogeny calculation is not clearly explained;
4. The tree in Fig. 3. shows numerous genera regarded as separate with ANI difference above 70% (cutoff value of the genus according to NCBI recommendations). Did the Authors consider the coverage values in the comparison?
5. Figure 5 is hardly readable, and Fig. 6 is unreadable in the presented pdf format.
Minor (technical) notes:
Line 25: differeNt
Line 33: G(capital)ram-negative
Lines 61-62: The statement of the superior conservatism of P2-like genomes is arguable. For example, T7-like genomes (Autographiviridae) are even more conservative. It is worth to specify, which "other classes" are compared.
Line 125 - The name of the cultuivation strain is EV67 or just EV?
Line 127 - city of the location of the manufacturer?
Line 154 "all seven genera of Eganvirus genus" - probably, species?
Lines 203-208 and 214-215 - please, add definitions of "complete" and "incomplete" lysis to the Materials&Methods section
Line 240 - "similarly the similarity" looks odd. I'd recommend to use "in a like manner" or "furthermore" instead of "similarly"
References 16, 18, 46 - please align the citation style
Reviewer 3 Report
Comments and Suggestions for Authors
The manuscript entitled: “The Phylogenetic Relationships and Evolution of the genus Eganvirus (186-type) Yersinia pestis Bacteriophages”
The manuscript is extremely well written, with little to no typos detected, except for some units not being separated from their numerical values. The results and insights described by the manuscript are extremely relevant, in my point of view. In addition, its scientific soundness is robust. I just have minor comments, however, I would like to have access to the authors responses and revisions, thus I am forced to recommend major amendments:
Line 90, gravitational force unit is not correctly displayed as “xg”, but it should be a italicized g separated from the number: “9000 g”.
Line 91, what is the material of the filter membrane? Please include that information.
Line 94 to 96, in my opinion a brief description of the methodology used should be described.
Line 116, please separate the values form the numerical values. Please revise the entire manuscript thoroughly.
Line 191, instead of “71.54 nm (mean ± SD: 71.54 ± 7.97 nm)”, please consider writing: 71.54 ± 7.97 nm. The same for the tail size.
Figure 1 scale must be improved. As it is, in my opinion, is unreadable.
Line 214, “Dark blue indicates complete lysis, light blue indicates incomplete lysis”, can the authors please detail what do you consider a complete and an incomplete lysis?
Figure 4 caption is in my opinion not entirely clear. If I missed it beg your pardon, but to what corresponds genera Eganvirus A and Eganvirus B? Furthermore, why is Felsduovirus represented here? Please comment.
Line 309 to 311, “Our group also isolated some Y. pestis phages belong to the genus Peduovirus (such as GMG73) that can lyse S. sonnei, S. soxoniae and E. 310 coli (this paper is being submitted)”, I understand the authors good intentions. However, in my opinion, this work was not yet peer-reviewed, thus it should not be mentioned.
Round 2
Reviewer 1 Report
Comments and Suggestions for Authors
The revised version contains accession numbers, but the corresponding genomes are not available in the NCBI databases. Please provide the genomes as .gbk files in the supplementary materials.
Revised Figure 1 (TEM of MHG48 and MLG42) contains two identical images. Please make changes.
The revised Figure 3 looks nice, but it does not provide the information needed to classify phages. Please provide the ANI values (multiplied by coverage) of the newly isolated phages and their closest relatives. You state that the VIRIDIC heatmap (or table) is contained in the supplementary materials, but I could not find it. ANI multiplied by coverage should be >70% for the same genus and you should use the genomic sequences of the closest phage already officially classified by the ICTV.
Please improve the visualization. At least keep one style in the design of your drawings. On the same trees, both italic and regular font are found in the species name.
Line 208: Please replace "Caudovirales" with "Caudoviricetes".
Reviewer 3 Report
Comments and Suggestions for Authors
I would like to acknowledge the authors improvements.
Furthermore, and I beg your pardon for not being clear, when I asked for: "please separate the values form the numerical values. Please revise the entire manuscript thoroughly", was just for the cases when the authors wrote: "88nm" insetad of "88 nm" for example.
